# Role of Portion Size in the Context of a Healthy, Balanced Diet: A Case Study of European Countries

**DOI:** 10.3390/ijerph20065230

**Published:** 2023-03-22

**Authors:** Michele O. Carruba, Maurizio Ragni, Chiara Ruocco, Sofia Aliverti, Marco Silano, Andrea Amico, Concetta M. Vaccaro, Franca Marangoni, Alessandra Valerio, Andrea Poli, Enzo Nisoli

**Affiliations:** 1Center for Study and Research on Obesity, Department of Biomedical Technology and Translational Medicine, University of Milan, Via Vanvitelli, 32, 20129 Milan, Italy; michele.carruba@unimi.it (M.O.C.);; 2Nutrition Foundation of Italy, Viale Tunisia, 38, 20124 Milan, Italy; 3Department of Food Safety, Nutrition and Veterinary Public Health, Istituto Superiore di Sanità, Viale Regina Elena, 299, 00161 Rome, Italy; 4Health and Welfare Unit, Censis Foundation, Piazza di Novella, 2, 00199 Rome, Italy; 5Department of Molecular and Translational Medicine, Brescia University, Viale Europa, 11, 25123 Brescia, Italy

**Keywords:** body weight, dietary intake, food portion size, health and healthy eating, obesity, standard reference portions, European countries

## Abstract

Over the past decades, a generalised increase in food portion sizes has probably contributed to the growing global obesity epidemic. Increasing awareness of appropriate portion sizes could contribute to reversing this trend through better control of calorie intake. In this study, a comparison of standard portion sizes in European countries for various food categories shows a wide variability of their importance for food, nutrient, and energy consumption according to government and institutional websites. On the other hand, the overall averages appear to be largely in line with the values indicated by the Italian Society of Human Nutrition, which is the most comprehensive and detailed document among those evaluated. The exceptions are milk and yoghurt, for which the reference portions in Europe are generally higher, and vegetables and legumes, for which portions are smaller than those reported in the Italian document. Moreover, the portion sizes of staple foods (e.g., pasta and potatoes) vary according to different food traditions. It is reasonable to consider that the creation of harmonised standard reference portions common to the European countries, based on international guidelines and scientific evidence, would significantly contribute to consumers’ nutritional education and ability to make informed choices for a healthy diet.

## 1. Introduction

A portion is generally defined as the amount of food people intend to consume on one eating occasion [1]. In its generic meaning as the “quantity or allowance of food allotted to, or enough for, one person” (Oxford English Dictionary, 2023), food portion may, therefore, not coincide (and be greater or smaller) with the so-called standard reference portion generally defined by experts or institutions at the national level [2]. For example, in Italy, a portion is defined as “the quantity of a food that is assumed to be a reference unit recognised and identifiable by both nutritional professionals and the general public” [3]; therefore, in keeping with dietary tradition, it should be of a reasonable size that follows consumer expectations and be expressed in units of measurement that refer to natural or commercial units or common household units. Standard portions are the references for the quantities of different foods to be recommended as part of diets for various age groups or groups with specific nutritional needs (e.g., pregnancy and breastfeeding) and to be indicated on the nutritional labelling of food products (such as those defined by the Academy of Nutrition and Dietetics in the United States). They are usually defined according to specific criteria, such as the history of use, product density, and intake data [4,5].

In recent years, the concept of portion size has been considered central by national (e.g., the Italian Consiglio per la Ricerca in Agricoltura e l’Analisi dell’Economia Agraria and the National Health Institute) and international organisations (e.g., World Health Organisation) [6,7,8], which include it among the determinants of dietary balance. Recently with co-funding from the European Union, the British Nutrition Foundation published a guide dedicated to portions, specifying how a healthy and balanced diet is determined by the choice of foods that make up such a diet and the appropriate quantities of the foods themselves [9].

The graphical representation of the Mediterranean Diet—a dietary pattern that a growing body of evidence confirms to be associated with a lower prevalence of chronic degenerative diseases and related risk factors [10]—is also based on the concept of portion size, as well as the frequency and variety of different food categories [11]. In particular, “moderation in portion size” is emphasised as a determinant for adjusting dietary intake to fit the specific needs of persons in modern societies, which are characterised by increasingly sedentary lifestyles, as well as for eliminating waste, thus favouring the sustainability of dietary patterns [12,13]. According to the of the Academy of Nutrition and Dietetics guidelines, no product is to be excluded if consumed in moderation and with adequate quantities as part of an overall balanced diet, thereby emphasising variety and moderation in the context of a healthy lifestyle to help reduce consumer confusion [14]. In fact, energy intake results from the size of the portions consumed and the frequency of consumption itself (i.e., the number of consumption occasions in a given period of time) [15]. An increase in one of these two factors, if not adequately compensated by a reduction in the other, can lead to an altered calorie intake.

Extensive literature has focused on the association between portion size and overweight/obesity, from the first ecological studies in the 1970s to the present. In the United States, for example, a progressive increase in the portion sizes of specific foods distributed in fast food chains and restaurants over thirty years (1977 to 2006) is considered to be responsible for an increase in energy intake, with some cases exceeding 100 kcal per unit of sale [16,17]. This phenomenon has been associated with a concomitant increase in the prevalence of overweight and obesity in the US population over the same period [18,19]. Similar associations have also been described in British children [20] and European adolescents [21].

Increased awareness of portion sizes could, therefore, play an essential educational role in promoting healthier eating and improving the consumption of certain foods, such as fruits, vegetables, and pulses, that have emerged as the determinants of the health benefits derived from specific eating patterns (e.g., the Mediterranean diet)—as already evident in some observational and intervention studies [11,22,23,24]. Our present analysis aims to conduct a comparative examination of standard portion claims for different food categories in European countries, which is crucial for a critical assessment of the possibility of defining harmonised portions to be adopted in different areas, ranging from possible nutrition policies to front-of-pack labelling proposals and food education strategies.

## 2. Materials and Methods

The information included in this analysis was obtained from documents published in various Member States of the WHO European Region by governmental bodies or scientific societies available online [25]. In particular, an extensive web interrogation was carried out to find institutional documents reporting information on portion sizes that apply to the adult population as the reference standards by using selected keywords translated from different European languages, such as “nutrition/food-based guidelines/recommendations”, “food policy”, “portion”, and “reference/standard portion”. We used the conversion table published by the Italian Society of Human Nutrition (available at https://sinu.it/wp-content/uploads/2019/07/20141111_LARN_Porzioni.pdf, accessed on 20 October 2022) to translate the various units of measurement (e.g., slices or cups) as they are indicated in the documents of the different countries analysed into a standard unit of measurement (i.e., grams, units, or millilitres). Since our study aimed to exclusively describe and highlight differences in portion sizes compared to the average, we did not consider assessing or evaluating their statistical significance necessary. This will be the aim of additional analysis of these data that will be elaborated on in a forthcoming manuscript.

## 3. Results

### 3.1. Standard Portion Sizes in EU Countries: The General Situation

Table 1 lists the 34 European countries for which relevant public documents (e.g., guidelines and government recommendations) were obtained as described. Critical issues found in the comparative evaluation include the reference for some countries pertains to the serving rather than to the portion (see, for example, Portugal) and, explicitly concerning grains (pasta and rice), the different use of grains in the preparation of meals (i.e., main course in some cases and accompaniment of dishes in others), which can lead to substantial differences in portion sizes.

All available data for the different food categories are listed in Table 2a,b for plant- and animal-based foods, respectively. The values expressed for different units, such as slices, cups, and spoons, have been reported in grams or millilitres where possible.

The average values calculated for all 34 countries and separately for the 24 EU and 10 non-EU countries are shown in Table 3. A comparison was also performed with the reference portion sizes, which have been defined by the Italian Society of Human Nutrition in the “Reference intake levels of Nutrients and energy for the Italian population” (LARNs) as a valuable tool for nutritional research and dietary planning [3].

The sometimes considerable difference between the minimum and maximum values recorded for the portion sizes in the countries reflects the variability in the portions within the same food category. However, a comparison of the median values calculated for the EU and non-EU countries confirms the existence of some homogeneity between the two groups of countries. A comparison with the standard reference portion sizes defined by the Italian Society of Human Nutrition also shows that, in most cases, the average values, both total and separately for the EU and non-EU countries, do not differ significantly from the corresponding Italian figures [3]. The exceptions are nuts, potatoes, vegetables, fish, legumes, and fresh cheese, for which the average portion sizes in the analysed countries are lower than the Italian reference portions, and milk and breakfast cereals, for which the standard portion sizes are generally higher in other countries than in Italy.

Analysing the average portion sizes across the countries according to geographical location (Southern, Central, and Northern Europe) allows us to appreciate a certain homogeneity of the data (Figure 1). The Italian reference values are higher for vegetables (200 g vs. 150 g on average, regardless of geographical area), fish and legumes (150 g for the Italian portion for each food category vs. an average value of just over 100 g for each food category for the other countries), and much smaller for a single portion of milk, which corresponds to a small glass (125 mL) in Italy, in comparison to an average of more than 200 mL (almost a large cup) in the other countries (Figure 1). Appendix A show the portion sizes measured for the different food categories in the diverse countries compared with the relative average values and the reference portion established by the Italian Society of Human Nutrition.

### 3.2. Differences and Similarities in Foods Portions in European Countries

According to the available documents, the amount of bread corresponding to one portion (50 g in Italy and seven other countries, including France, Spain, Austria, Germany, and Finland) varies from 15 g in Slovenia to 100 g in Iceland, Serbia, and Switzerland (Appendix A).

For “Pasta, rice, maize, barley, spelt”, the generic portion size indicated is 80 g for Italy, as well as for Germany, Malta, Norway, Spain, and Sweden (Appendix A). Larger portions (above 100 g) are suggested in Belgium and Turkey; similarly, Cyprus, Hungary, Lithuania, and Slovakia adopt a portion of 100 g. In other countries, the portions are smaller than the average: 50 g in Estonia, 35 g in Portugal, and 30–35 g for pasta and 20 g for rice in Croatia. Interestingly, for some countries, the portion also refers to the cooked product: 110 g for Portugal, 70 g for Estonia, 200 g for Hungary, 125 g for the Czech Republic, and 150 g for Lithuania.

Average portion sizes for ready-to-eat breakfast cereals and potatoes are 39 g and 152 g, respectively (Appendix A).

For fresh fruits, a portion size of about 150 g prevails in 12 countries, as in Italy, which also coincides with the general average (Appendix A). In the Czech Republic, Estonia, Hungary, and Norway, the reference portion is set at 100 g, slightly higher than the 80 g suggested in Malta, Slovenia, and the United Kingdom. In Belgium and Germany, the portions are considerably higher than the average (250 g).

For seven countries, two reference portions prevail for vegetables and greens: 100 g and 200 g. The average size is nearly 160 g, ranging from 80 g in Malta, Poland, Slovenia, and the United Kingdom to 300 g in Armenia and Hungary (Appendix A). Concerning leafy vegetables, the standard portions are generally lower, at 80 g in Italy and ranging from 50 g in Lithuania to 200 g in Croatia.

The portion size for fresh or canned pulses (Appendix A) is 150 g in Italy, as well as in Austria, Greece, and Ireland, compared to an overall average of just over 110 g and a peak of 200 g in Lithuania. The portion size of dried legumes, which in Italy is defined as 50 g, is also reported in Estonia (10 g), Portugal and Croatia (25 g), Poland (40–60 g), Spain (60–80 g), Malta (70 g), and finally Slovenia (4 tablespoons or 50 g) (Appendix A).

For red meat, the 100 g stated by the LARNs for the reference portion size is shared by Armenia, Hungary, Iceland, the Netherlands, North Macedonia, Switzerland, and Turkey and is very close to the overall average value, which ranges from 150 g in Norway, Poland, and Slovenia to 30 g in Portugal, Cyprus, and Croatia (Appendix A). For white meat, the portion sizes remain the same as for red meat in Italy, Malta, Portugal, Hungary, Cyprus, Poland, the Czech Republic, and Croatia (100 g), which coincides with the overall average (Appendix A).

The portion size suggested for fish is sometimes larger than for meat, not only in Italy (150 g, as well as in Greece, Hungary, Austria, and Poland) but especially in Norway (175 g) and Spain (180 g) (Appendix A). The overall average is about 110 g.

The reference portion for eggs corresponds to one egg in most countries.

A fair variability is observed for the reference portions of milk (Appendix A): the smallest, corresponding to 125 mL, is the one defined in Italy (i.e., one glass), as well as in Belgium and Lithuania. In other countries, larger portion sizes prevail (one “cup” or similar, corresponding to 200–250 g); consequently, the overall average is around 210 mL. Additionally, for yogurt, the 125 g indicated by the LARNs in Italy corresponds to the smallest portion, with an overall average of around 180 g (Appendix A).

The portions of cheeses, separated in most cases into fresh and hard cheeses (soft and hard), show considerable differences, with the averages being below the LARN values in both cases (Appendix A): 75 g against 100 g for the leanest cheeses and just over 35 g against 50 g for the most aged cheeses.

Concerning vegetable oils, including olive oil, the standard portion size is 10 g, not only in Italy but also in Albania, France, Georgia, Germany, Portugal, Serbia, Spain, Sweden, and Switzerland (Appendix A).

In Italy, standard portions are also defined for sweet food products: “cakes, spoon sweets, and ice cream”, “snacks, crisps and chocolate bars”, “sugar”, and “honey and jam” (for the latter two categories, the portions are also defined in Hungary and Estonia).

More scattered are the reference portion values for beverages. A single portion of water is 200 mL in Italy, Spain, and Portugal; between 200 and 250 mL in Hungary; and 250 mL in Malta, Greece, and Slovenia. For fruit juices, iced tea, and soft drinks, the standard portion size is 200 mL (330 mL if in a can) in Italy, 100 mL in Estonia, 250 mL in Hungary (200 mL if in a can), 150 mL in Spain, and 125 mL in Greece. Finally, the consumption unit for wine is 125 mL in Italy and Greece, 100 mL in Slovenia and Spain, and 80 mL in Malta. The reference portion rises to 330 mL for beer in Italy and Greece, 250 mL in Slovenia and Malta, and 200 mL in Spain and Portugal. For spirits, the portion size is smaller: 25 mL in Malta, 30 mL in Slovenia, 40 mL in Italy, and 40–45 mL in Greece.

## 4. Discussion

The analysis of the documents available on government and institutional websites shows how attention to the concept of portion size, as a determining element of food, nutrient, and energy consumption, is heterogeneous in the countries considered in this study. On the other hand, the evaluation of the overall averages reveals a substantial overlap in most cases with the values indicated by the LARNs as defined by the Italian Society of Human Nutrition, which represents the most detailed and complete document among those examined [3]. The exceptions are the cases of milk and yoghurt, for which most of the reference portions in Europe are larger than in Italy, and the portions of vegetables and legumes are smaller than those reported in the Italian document.

In general, portions of foods that are more likely to be portioned per se, such as fruits, potatoes, and fish, tend to be more homogeneous than liquids or foods that are more difficult to refer to in terms of consumption units, such as milk and cheese. Another general observation concerns portion sizes, which are larger overall for staple foods (such as pasta and potatoes) and, therefore, vary according to the food traditions of the different countries: the same portion of pasta has a higher weight if it is considered a staple food than in countries where it is simply an accompaniment to dishes.

Previous research has reported the need for more documents focusing on reference portion sizes in other EU countries [59,60]. However, the need to convey to the population indications regarding actual food and drink intakes stems mainly from the observation that larger portions encourage food intake and, if this is in excess, contribute to an increase in calorie intake and, consequently, in the risk of the onset of overweight and obesity [61].

The possible role of portions in determining food consumption quantities has been confirmed by a Cochrane Review [62]. The authors proposed that the size of tableware used at home should also be reduced to help improve food choices and consumption in quantitative terms. Based on this review’s results, however, reducing portion sizes does not appear to be a uniformly effective strategy: the available data show that reducing portions at the larger end of the size range leads to a reduction in food intake but does not allow a recommendation regarding whether reducing portions at the smaller end of the size range would be equally effective.

Interestingly, a clear and significant association between portion sizes in school canteens and the risk of excess weight was described in a study conducted on a population of Italian children: the obesity rate was higher among children exposed to larger portions that were further away from the standard recommended portion sizes for children of the same age [63]. Confirmation of the role of portion sizes in overall food intake comes from a meta-analysis of 58 studies (with a total of 6603 participants): the effect that portion size, packaging, single units, or household tableware can have on the food consumption levels of adults and children, although statistically small or moderate, is significant. This suggests that controlling consumption units and, thus, portions, both at home and when outside, could effectively limit average energy intake by a relatively large proportion (about 10% on average, according to the authors) [62]. On the other hand, it has been hypothesised that the inability to recognise the amount of food consumed on a single occasion may be an obstacle to controlling food intake [64]. The effect of the failure to regulate food intake with prolonged exposure to larger portions has been described in preschool children, particularly those with higher body weights, challenging the assumption that self-regulatory behaviour would be sufficient to counteract perturbations in energy intake [65]. Several factors can modulate this portion size effect [66]. This effect is particularly evident in the conditions in which the size of the portions directly influences the quantity of food consumed, as in the case of the Italian school canteens discussed above.

A similar effect, although probably less marked in terms of quantity, can also be hypothesised for industrial foods presented in a clearly portioned form. Sharing the concept of standardised portions with food companies could, therefore, expose consumers to more appropriate amounts of packaged foods by helping companies correctly recognise the quantities of food to be consumed on a single occasion. This alliance between institutions and food companies should be vigorously pursued.

On the other hand, the size of the sales unit is one of many variables to be considered. More variable and subjective factors also play a role in determining this effect, which is linked to both the individual and his/her context: for example, in physiological conditions, people who eat more quickly or have difficulty perceiving a sense of satiety are inclined to consume larger portions [67]. Furthermore, eating in the company of others or alone can condition the portions of food or drink consumed.

The aforementioned Italian study highlighted the role of exposure to adequate portion sizes for children, who were more likely to be of normal weight if they had access to school canteens that provided meals quantitatively in line with nutritional recommendations [63]. On the other hand, it is clear that when a consumer decides autonomously the quantity of food to consume and does not find “ready-made” portions as in the school canteens mentioned above, the information in his/her possession becomes decisive.

It is, in fact, undeniable that knowledge of the actual portion size can contribute to making consumption choices that are more balanced and conscious. A survey of more than 13,000 people in 6 European countries (Germany, United Kingdom, Spain, France, Poland, and Sweden) showed that consumers with a higher focus on health-related topics were also those who found information about the amount of product per portion most relevant [68]. Furthermore, almost all respondents in the various countries agreed on the definition of portion as the amount of food that should be consumed; only the respondents in Sweden identified the amount of food they could eat as a portion. By using a sequence of ad hoc distance tests, this study established how the availability of nutritional information on product packaging per portion, as well as per 100 g or 100 mL, could help consumers make more informed food choices: in fact, the percentage of respondents who were able to identify the amount of nutrients and energy per portion increased significantly (for saturated fat, for example, from 15% to 86%), and in a much shorter time, moving from the situation where the presence of nutritional composition was only referred to as per 100 g of product to one where the same information was reported per portion [69].

A large-scale communication and information campaign on the concept of portions could, therefore, greatly help improve the quantitative aspects of the general population’s food consumption. In this regard, the use of front-of-pack labelling, which is being studied in the European community, could also contribute, provided that it is based on the adequate portion to be consumed. Such an approach would also make it possible to integrate this quantitative information with information on the qualitative composition of foods, thus facilitating the choices necessary to follow a balanced and complete food pattern.

### Standard Portions vs. Consumption Units

The potential importance of the definition of standard reference portions to be used on nutrition labels in the European Union was pointed out by the authors of the Food4Me study, a multi-center, web-based protocol to determine the effects of personalised versus conventional nutrition communication to the general population [60]. The analysis of the food choices of selected population groups in Germany, Greece, Ireland, the Netherlands, Poland, Spain, and the United Kingdom revealed an overall homogeneity of the portions consumed in each country when compared to the national averages, but also a difference between the average portions in use in the different countries, which was significant for 42% of the 156 foods considered from various categories. The most significant differences were in the categories of “grains”, “potatoes”, “rice and pasta”, and “meat and fish”; less heterogeneous were the consumption unit sizes for “soups”, “sauces and condiments”, “fats and spreads”, and “fruit”. The average portion sizes consumed differed from the weighted average for the study population in all seven countries in only 15.7% of the cases, suggesting that attention should, therefore, be paid to those products which portion sizes, in general, differed significantly from the weighted average, and to those countries (mainly Ireland among the countries in the study) where portion sizes differed significantly more than the average. Acknowledging the lack of standardised portion sizes common to European countries at the time of publication, the authors themselves admitted that the typical portion sizes in use in different countries might even differ significantly from the 100 g or 100 mL indicated by the European legislation, pointing out that the harmonisation of standard portions is of potentially great value for the definition of reference quantities based on which nutritional information can be provided, which is in agreement with the results of previous research [69,70].

This aspect appears to be crucial: information on the nutritional composition of individual portions or sales/consumption units of a product is recognised as valuable support for consumers, who are, thus, able to immediately understand the amount of energy and nutrients obtained with the consumption of that food (Food Standards Australia and New Zeland) [71]. On the other hand, consumer confusion regarding the concept of adequate portion sizes for different foods is clear from the available literature [2]. Indeed, the authors of a review of the results of five research studies emphasise the importance of providing accurate information on the size of different portion sizes in line with the recommendations for healthy eating that are fundamental for proper nutrition education [72].

The appropriateness of standardisation of reference portion sizes based on nutritional guidelines also emerges strongly from the findings of an analysis of food products for sale in the Australian market and a review of available literature aiming to create an effective strategy to promote the production of more appropriate sales/consumption units by companies [73], to provide consumers with the information they need to align intake levels with recommendations [11], and to improve the quality of nutritional information on packaged product labels [19,74].

The observation that has emerged from the available studies points out that the definition of standardised and harmonised portions in the European Union could advance the definition of nutritional recommendations and, therefore, promote educational programmes on healthy eating that are common to various countries, while respecting the different eating habits and traditions and facilitate the communication of nutritional information by referring to the quantity of product consumed (rather than to 100 g or 100 mL). This could have a twofold result: educating people to consume different foods in appropriate quantities that are compatible with a balanced diet and helping them understand the contribution of foods to the overall diet [1,75]. Finally, the definition of reference food portions appears to be a proper strategy to improve the quality of diet and, thus, consumer well-being and reduce food waste and its associated costs [76].

## 5. Conclusions

In conclusion, the dissemination of the concept of portion sizes for various foods and the definition, based on international guidelines and scientific evidence, of harmonised standard reference portions common to the countries of the EU could make a significant contribution to the improvement of nutritional information provided to consumers and, therefore, enhance their ability to make informed choices for an overall healthy diet. Portion sizes can differ significantly depending on the country or culture. Cultural attitudes towards food, cuisine, eating habits, and food availability and affordability can influence portion sizes. It is important to note that portion sizes can also vary within a country based on regional differences, socioeconomic status, and personal preferences. However, an analysis of the reference portions available in the European countries to date suggests that the creation of harmonised standard portions for the main categories of food is desirable and decidedly practicable, following the example of countries that have made progress in this direction, such as Italy. However, particular attention must be paid to selected food categories, such as vegetables, legumes, fish, milk, and derivatives, for which the available scientific evidence supports the promotion of consumption that can easily be achieved by indicating more consistent reference portions in line with needs. Based on these considerations, the indication of harmonised standard reference portions appears relevant, especially for the definition of simplified front-of-pack labelling systems to optimise the transmission of correct information to consumers to allow them to contextualise the consumption of foods with different nutritional characteristics in an overall balanced diet.

## Figures and Tables

**Figure 1 ijerph-20-05230-f001:**
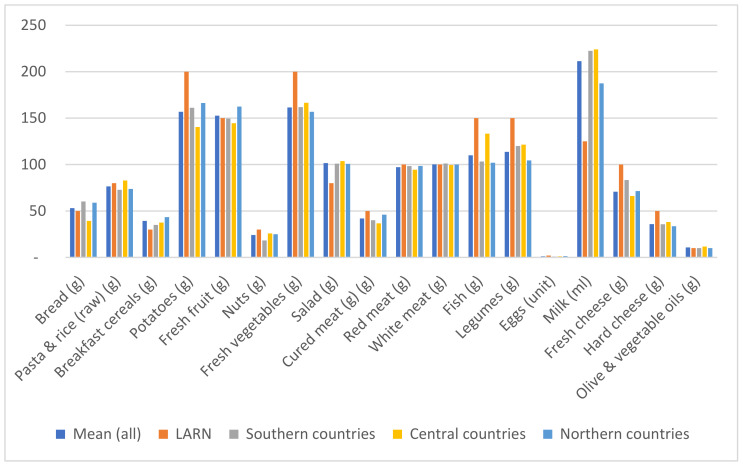
Mean portion sizes in the considered countries (overall mean; Southern, Central, and Northern countries) and reference standard portions defined by the Italian Society of Human Nutrition (LARNs, “Reference intake levels of Nutrients and energy for the Italian population”) [3].

**Table 1 ijerph-20-05230-t001:** Standard portions in 34 European countries: reference documents.

Country	Reference Document(s)	Year	Ref.
Albania	Recommendations on Healthy Nutrition in Albania	2008	[26]
Armenia	Food-Based Dietary Guidelines in the WHO European Region	2003	[27]
Austria	The Austrian Food Pyramid—7 Steps to Health	2010	[28]
Belgium	Dietary Guidelines for the Belgian Adult Population	2019	[29]
Croatia	Dietary Guidelines for Adults	2002	[30]
Cyprus	Nutrition and Exercise Guidelines	2007	[31]
Czech Republic	Nutritional Recommendations for the Population of the Czech Republic	2012	[32]
Denmark	The Official Dietary Guidelines—Good for Health and Climate	2021	[33]
Estonia	Estonian Recommendations for Nutrition and Exercise	2015	[34]
Finland	Nutrition and Food Recommendations	2023	[35]
France	Actualisation des repères du PNNS: révision des repères de consommations alimentaires	2016	[36]
Georgia	Health Eating—the Main Key to Health	2005	[37]
Germany	The DGE Nutrition Circle	2011	[38]
Greece	National Dietary Guide for Adults	2014	[39]
Hungary	Dietary Guidelines for the Adult Population in Hungary	2004	[40]
Iceland	Dietary Recommendations for Adults and Children from Two Years of Age	2016	[41]
Ireland	Healthy Eating, Food Safety and Food Legislation—A Guide Supporting the Healthy Ireland Food Pyramid	2011	[42]
Italy	LARN—Reference Intake Levels of Nutrients and Energy for the Italian Population	2014	[3]
Latvia	Nutrition Recommendations for Adults	2020	[43]
Lithuania	Recommendations for a Healthy and Sustainable Diet	2020	[44]
Malta	Dietary Guidelines for Maltese adults: Healthy Eating—The Mediterranean Way	2015	[45]
Netherlands	Dutch Dietary Guidelines	2015	[46]
Northern Macedonia	Dietary Guidelines	2013	[47]
Norway	Nordic Nutrition Recommendations 2012 Integrating Nutrition and Physical Activity	2014	[48]
Poland	Nutritional Recommendation for the Polish Population and Their Application	2020	[49]
Portugal	The New Food Wheel, a Guide to a Daily Food Choice	2003	[50]
Serbia	Improving Nutrition Surveillance and Public Health Research in Central and Eastern Europe/Balkan Countries Using the Balkan Food Platform and Dietary Tools	2008	[51]
Slovakia	Ten Rules of a Healthy Plate	2003	[52]
Slovenia	Healthy Plate: Recommendations for Healthy Eating	2010	[53]
Spain	Guide to Healthy Eating	2019	[54]
Sweden	The Swedish Dietary Guidelines	2002	[55]
Switzerland	The Swiss Food Pyramid—Dietary Recommendations for Adults, Combining Pleasure and Balance	2007	[56]
Turkey	Dietary Guidelines for Turkey	2012	[57]
United Kingdom	Easy to Eat Well	2021	[58]

**Table 2 ijerph-20-05230-t002:** (**a**). Portion sizes according to the published documents for different countries: plant-based foods. Green: values equal to the median; blue: values below the median; and red: values higher than the median. Legumes are reported as a portion of fresh, soaked, frozen, or canned legumes. (**b**). Portion sizes according to the published documents for different countries: animal-based foods. Green: values equal to the median; blue: values below the median; and red: values higher than the median.

(a)										
Country	Bread (g)	Pasta and Rice (Raw) (g)	Breakfast Cereals (g)	Potatoes (g)	Fresh Fruits (g)	Nuts (g)	Fresh Vegetables (g)	Salad (g)	Olive and Vegetable Oils (g)	Legumes (g)
Albania	50	--	50	188	175	30	125	85	10	120
Armenia	-	50		250	200	-	300	-	-	-
Austria	50	70	50	200	150	20	250	80	15	150
Belgium	-	125	-	-	250	30	-	-	-	100
Croatia	30	30	20	100	150	10	150	200	5	100
Cyprus	30	100	-	90	150	-	200	-	5	100
Czech Republic	60	70	-	-	100	-	-	-	-	-
Denmark	75	60	-	140	200	30	200	-	-	100
Estonia	30	50	20	100	100	10	100	125	5	30
Finland	50	50	-	-	150	15	150	150	-	100
France	50	-	-	-	150	20	100	80	10	100
Georgia	60	-	-	100	125	-	100	-	10	190
Germany	50	85	50	200	250	25	200	100	10	125
Greece	30	80	30	100	150	-	200	-	15	150
Hungary	80	100	40	200	100	-	100	100	-	-
Iceland	50	-	-	200	200	-	300	-	-	100
Ireland	100	100	50	200	150	40	100	80	5	150
Italy	50	80	30	200	150	30	200	80	10	150
Latvia	70	75	-	100	150	-	150	80	-	-
Lithuania	28	100	40	100	150	-	100	50	-	200
Malta	28	80	40	80	80	20	80	-	15	140
Netherlands	35	90	50	70	200	20	250	-	15	60
N. Macedonia	-	-	-	-	220	-	220	-	-	-
Norway	40	80	-	-	100	30	100	100	-	-
Poland	90	-	-	-	-	-	80	-	15	-
Portugal	50	35	35	125	160	-	180	-	10	80
Serbia	100	70	-	260	150	30	175	85	10	130
Slovakia	70	100	40	-	200	-	200	-	-	-
Slovenia	14	34	24	100	80	-	80	80	-	50
Spain	50	80	-	200	150	-	200	150	10	-
Sweden	40	80	40		200	30	250	-	10	-
Switzerland	100	60	60	200	120	-	120	-	10	-
Turkey	25	130	-	-	130	15	130	-	20	110
UK	60	-	-	-	80	30	80	-	-	80
Median	50	80	40	140	150	28	150	85	10	100
(**b**)										
**Country**	**Cured Meat** **(g)**		**Red Meat** **(g)**	**White Meat** **(g)**	**Fish** **(g)**	**Legumes (g)**	**Eggs (Unit)**	**Milk (mL)**	**Fresh Cheese (g)**	**Hard Cheese (g)**
Albania	30		120	150	-	120	2	250	50	45
Armenia	-		100	100	30	-	-	200	-	35
Austria	-		125	-	150	150	1	200	100	-
Belgium	-		125	125	100	100	-	125	-	-
Croatia	60		30	30	30	100	-	200	100	30
Cyprus	30		30	30		100	1	240	30	30
Czech Republic	-		125	125	125	-	-	250	-	-
Denmark	-		-	-	-	100	-	250	-	-
Estonia	40		35	60	70	30	1	200	100	40
Finland	-		125	125	125	100	-	-	-	-
France	50		-	130	100	100	1	150	45	30
Georgia	-		80	80	80	190	1	250	-	30
Germany	20		125	125	100	125	1	200	-	30
Greece	-		135	135	150	150	1	250	60	30
Hungary	40		100	100	150	-	1	200	50	30
Iceland	-		100	100	100	100	-	250	-	25
Ireland	-		60	60	100	150	2	200	25	25
Italy	50		100	100	150	150	1	125	100	50
Latvia	-		-	-	-	-	-	250	-	-
Lithuania	-		70	70	100	200	1	125	100	40
Malta	-		90	100	115	140	1	250	50	40
Netherlands	-		100	100	100	60	1	150	-	40
N. Macedonia	-		100	100	-	-	-	-	-	-
Norway	-		150	150	175	-	-	200	100	20
Poland	-		150	150	150	-	-	-	-	-
Portugal	-		30	30	30	80	1	250	75	40
Serbia	30		-	130	-	130	2	250	40	30
Slovakia	-		-	-	-	-	-	250	-	50
Slovenia	-		150	-	100	50	1	200	40	-
Spain	-		-	100	180	-	1	250	100	50
Sweden	-		-	-	-	-		250	-	-
Switzerland	-		100	100	100	-	2	200	150	50
Turkey	-		100	100	-	110	-	175	30	-
UK	70		70	-	140	80	-	-	-	-
Median	40		100	100	100	100	1	200	60	33

**Table 3 ijerph-20-05230-t003:** Portion sizes for the different food categories: average, minimum, maximum, and median values in all countries analysed (n = 34), and median values calculated separately for European Union (EU) (n = 24) and non-EU (n = 10) countries, in comparison with the portion sizes defined by the LARNs (Reference intake levels of nutrients and energy for the Italian population) [3].

	Mean All (n = 34)	Min	MAX	Median All (n = 34)	Median EU (n = 24)	Median Non-EU (n = 10)	LARN [3]
Bread (g)	53	14	100	50	50	55	50
Pasta and rice (raw) (g)	76	30	130	80	80	70	80
Breakfast cereals (g)	39	20	60	40	40	55	30
Potatoes (g)	152	70	260	140	100	200	200
Fresh fruits (g)	154	80	250	150	150	140	150
Nuts (g)	24	10	40	27,5	20	30	30
Fresh vegetables (g)	162	80	300	150	165	127	200
Salad (g)	102	50	200	85	80	85	80
Olive and vegetable oils (g)	11	5	20	10	10	10	10
Cured meat (g)	42	20	70	40	40	30	50
Red meat (g)	97	30	150	100	100	100	100
White meat (g)	100	30	150	100	100	100	100
Fish (g)	110	30	180	100	100	100	150
Legumes (g)	114	30	200	100	100	115	150
Eggs (unit)	1	1	2	1	1	2	2
Milk (mL)	211	125	250	200	200	225	125
Fresh cheese (g)	71	25	150	60	67	50	100
Hard cheese (g)	36	20	50	32	40	30	50

## Data Availability

Data are contained within the article or Appendix A.

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
