# Peer review of "Role of Portion Size in the Context of a Healthy, Balanced Diet: A Case Study of European Countries"

_ijerph, 2023, doi:10.3390/ijerph20065230_

Round 1

Author Response

Reviewer 1

This paper addresses the problem of the importance of portion sizes, and of their lack of uniformity in the various European countries. While the Results section is boring because of the detailed description of the differences in portions between countries, which can be presented in a table and not detailed in the text, the discussion is well conducted and focuses on the main reasons that make portion size important, exposing the current literature on this topic.

I suggest that authors perform a major revision of the Results section, making it more simply thanks to the reference to tabulated data, and highlighting only the important points in the text.

We thank this Reviewer for his/her overall favourable judgment of our manuscript. We have done our best to revise the Results section as suggested.

Original: Cereals (in the text); Suggested: If referred to pasta, bread, etc., it’s better to use “grains”

O.K., we have changed the text as requested (please, see lines 110, 111, and 329 of the revised version of the manuscript).

Original: Caloric (in the texts); Suggested: calorie

O.K., we have changed the text as requested (please, see lines 17 and 249 of the revised version of the manuscript).

Original: Table 2, figure 1; Suggested: It’s suggested a footnote to explain the meaning of LARN. Legumes fresh/cooked?

O.K., we have added the requested information (for LARNs, please, see lines 128-130, 133-134, and 159-160; for legumes, please see line 120 of the revised version of the manuscript).

Original: LARN (in the text); Suggested: Or LARNs? Use the two wit congruence.

We have changed into LARNs as suggested (please, see the previous reply for the changed lines of the revised version of the manuscript).

Original: Subparagraphs from 3.2 to 3.7; Suggested: The detailed description of the portion size in the different countries is boring and can be substituted by table 1S (in an improved formatting), and in the text can be commented only relevant aspects, not suitable for the table.

We fully agree with and thank this Reviewer for his/her suggestion. In particular, we have replaced a part of the Results section with the previous Supplementary Table 1 (now mentioned as Table 2a and Table 2b, reporting portion sizes of plant- and animal-based foods, respectively, in the revised version of the manuscript).

Reviewer 2 Report

The manuscript on “Role of the portion size in the context of a healthy, balanced diet” can be a valuable contribution to the field of nutrition and public health. The aim of the analysis presented within the paper is the comparative examination of standard portion claims for different food categories in European countries, underlying the role of portion sizes in achieving a healthy and balanced diet, which is a crucial topic given the increasing prevalence of diet-related diseases worldwide. The manuscript discusses the variations in portion sizes across different countries, which is an important consideration for designing effective public health interventions. However, it is important to note that the manuscript could benefit from certain improvements, particularly as outlined below:

1)      I would like to suggest editing the title of the paper as follows: “Role of the portion size in the context of a healthy, balanced diet: A case study of European countries." This is because the research only provides data from European countries, and the discussions are centered around these cases. The current title "Role of the portion size in the context of a healthy, balanced diet" is too generic and does not accurately reflect the focus of the study.

2)      Another suggestion is to enhance the “Materials and Methods” section by providing more clear and detailed information on the methods used, particularly including an explanation of the statistical analyses used to evaluate the homogeneity of portion size data, or the statistical test used to determine significant differences in mean portion sizes across multiple countries when comparing data between different countries. This would help ensure greater clarity and transparency regarding the research methods implemented.

3)      Please provide clarification on whether the information regarding “Standard portions in different European countries” is applicable to the adult population, or if it pertains to both adults and children.

4)      Please consider these concluding remarks in the section “Conclusion”: “Portion sizes can differ significantly depending on the country or culture. Cultural attitudes towards food, cuisine, and eating habits, as well as food availability and affordability, can influence portion sizes. It’s important to note that portion sizes can also vary within a country based on regional differences, socio-economic status, and personal preferences.”

5)      Please provide the full name of “LARN” (Revisione dei Livelli di Assunzione di Riferimento di Nutrienti ed energia per la popolazione italiana) or a brief explanation with a reference citation as a note after all tables or figure that include this abbreviation. This would ensure that readers who may not be familiar with this term can understand its meaning and context within the study.

6)      In “Table 1. Standard portions in different European countries: reference documents.” It would be better to write the reference document’s title and the relevant citation (i.e. number as in the reference list). Please, pay attention that all the provided links in the table should be properly included in the reference list. For example, FAO’s document for Albania presented in the table is not listed in the “Reference”.

7)      Please review “Supplementary Table 1: Portion sizes according to published documents for different countries.” It appears that certain numbers available in the corresponding references are missing in this table. For example, the portion size of bread for Armenia is not presented in the table, although it is provided in the WHO’s cited document on FBDG. Besides, for the missing data authors can write as “-” with a possible available explanation as a note after the table. Clarifying and updating the missing information in the table would enhance the accuracy in the study.

Author Response

Reviewer 2

The manuscript on “Role of the portion size in the context of a healthy, balanced diet” can be a valuable contribution to the field of nutrition and public health. The aim of the analysis presented within the paper is the comparative examination of standard portion claims for different food categories in European countries, underlying the role of portion sizes in achieving a healthy and balanced diet, which is a crucial topic given the increasing prevalence of diet-related diseases worldwide. The manuscript discusses the variations in portion sizes across different countries, which is an important consideration for designing effective public health interventions. However, it is important to note that the manuscript could benefit from certain improvements, particularly as outlined below:

We thank this Reviewer for his/her appreciation of our manuscript. We have taken into consideration all of the suggestions to improve the manuscript.

1) I would like to suggest editing the title of the paper as follows: “Role of the portion size in the context of a healthy, balanced diet: A case study of European countries." This is because the research only provides data from European countries, and the discussions are centered around these cases. The current title "Role of the portion size in the context of a healthy, balanced diet" is too generic and does not accurately reflect the focus of the study.

We agree and have changed the title of the paper as suggested.

2) Another suggestion is to enhance the “Materials and Methods” section by providing more clear and detailed information on the methods used, particularly including an explanation of the statistical analyses used to evaluate the homogeneity of portion size data, or the statistical test used to determine significant differences in mean portion sizes across multiple countries when comparing data between different countries. This would help ensure greater clarity and transparency regarding the research methods implemented.

We thank this Reviewer for his/her suggestion. We have improved the "Materials and Methods" section by providing more precise and detailed information on the method used to translate the various units of measurement - as they are indicated in the documents of the different countries analysed - into a standard unit of measurement (i.e. grams, units or millilitres). As now reported, "we used the conversion table published by the Italian Society of Human Nutrition (available at https://sinu.it/wp-content/uploads/2019/07/20141111_LARN_Porzioni.pdf) (please, see lines 96-100 of the revised version of the manuscript). Since our study aimed only to describe and highlight differences in portion sizes compared to the average, we did not consider assessing or evaluating their statistical significance necessary. We thank the Reviewer very much for suggesting a statistical analysis and reporting "the methods used to evaluate the homogeneity of portion size data [and] determine significant differences in mean portion sizes across multiple countries when comparing data between different countries." Such will be the aim of additional analysis of these data, to be elaborated in a forthcoming manuscript. As a result, we considered comparing individual country values with the mean value sufficient to identify and discuss in this paper the most obvious peculiarities, usually attributed to economic or cultural reasons. Nonetheless, we have explicitly outlined our position in the "Materials and Methods" section of the revised version of the manuscript (please, see lines 101-104 of the revised version of the manuscript), hoping that this description at least partially meets the Reviewer's requirements.

3) Please provide clarification on whether the information regarding “Standard portions in different European countries” is applicable to the adult population, or if it pertains to both adults and children.

We thank this Reviewer for the opportunity to clarify our approach. Portion sizes applicable to the adult population as reference standards are reported in the institutional documents (we have specified this point in lines 93-94 of the revised version of the manuscript). According to the Italian dietary guidelines, portion sizes may differ per children's age group.

4) Please consider these concluding remarks in the section “Conclusion”: “Portion sizes can differ significantly depending on the country or culture. Cultural attitudes towards food, cuisine, and eating habits, as well as food availability and affordability, can influence portion sizes. It’s important to note that portion sizes can also vary within a country based on regional differences, socio-economic status, and personal preferences.”

Thank you. We have written as suggested (please, see lines 376-379 of the revised version of the manuscript).

5) Please provide the full name of “LARN” (Revisione dei Livelli di Assunzione di Riferimento di Nutrienti ed energia per la popolazione italiana) or a brief explanation with a reference citation as a note after all tables or figure that include this abbreviation. This would ensure that readers who may not be familiar with this term can understand its meaning and context within the study.

We agree. We have provided the full name of “LARNs” as suggested in the revised version of the manuscript (please, see lines 128-130, 133-134, and 159-160).

6) In “Table 1. Standard portions in different European countries: reference documents.” It would be better to write the reference document’s title and the relevant citation (i.e. number as in the reference list). Please, pay attention that all the provided links in the table should be properly included in the reference list. For example, FAO’s document for Albania presented in the table is not listed in the “Reference”.

Thank you very much for these suggestions we applied in the revised version of the manuscript (please, see Table 1 and Reference 26).

7) Please review “Supplementary Table 1: Portion sizes according to published documents for different countries.” It was done (please, see lines 119 and 123 of the revised version of the manuscript).

It appears that certain numbers available in the corresponding references are missing in this table.  You are right: we have checked this more carefully. Thank you very much.

For example, the portion size of bread for Armenia is not presented in the table, although it is provided in the WHO’s cited document on FBDG. We have assumed that the amount reported for bread in the Armenian document, 250 g, corresponds to the total daily intake recommended for this food. Information was not found on the size of a single standard portion.

Besides, for the missing data authors can write as “-” with a possible available explanation as a note after the table. Clarifying and updating the missing information in the table would enhance the accuracy in the study. Thank you for the suggestion. We have selected countries where information on the standard portions is available for the different food categories. The number of food categories with a defined standard portion varies from 4 in North Macedonia to 18 in Italy and Estonia.

Reviewer 3 Report

The article presents a comparative analysis of standard portions of various categories of food in European countries and a critical assessment of the possibility of determining harmonized portions that will help ensure a rational diet for the population. The authors briefly and correctly review scientific data and legislative acts in the field of rational nutrition.

 The results of this study are quite interesting and useful for readers. However, there are some issues that need to be reconsidered.

It is necessary to specify in the Materials and Methods section which algorithm you have used for the data processing.

Table 2 – give explanations for the indicators: Mean (all);  LARN 

As a rule, when drawing up a personalized diet, more attention is paid to the energy value of the diet and the ratio of the main components (proteins, fats and carbohydrates) depending on their share (in %) of the caloric content of the diet.

To improve the article Scientific Soundness, I recommend adding a comparison of the portion size of different food categories with their caloric content (add a table or chart).

Author Response

Reviewer 3

The article presents a comparative analysis of standard portions of various categories of food in European countries and a critical assessment of the possibility of determining harmonized portions that will help ensure a rational diet for the population. The authors briefly and correctly review scientific data and legislative acts in the field of rational nutrition.

 The results of this study are quite interesting and useful for readers. However, there are some issues that need to be reconsidered.

Thank you for the Reviewer's overall appreciation of our manuscript.

It is necessary to specify in the Materials and Methods section which algorithm you have used for the data processing.

As specified in the Materials and Methods section (lines 92-96), we used “an extensive web interrogation […] to find institutional documents reporting information on portion sizes applicable to the adult population as reference standards by using selected keywords translated from different European languages, such as “nutrition/food-based guidelines/recommendations”, “food policy”, “portion”, “reference/standard portion”. No algorithm was used for the data processing.

Table 2 – give explanations for the indicators: Mean (all);  LARN 

We agree. We have provided the full name of “LARNs” as suggested in the revised version of the manuscript (please, see lines 122-124, 127-128, and 153-154).

As a rule, when drawing up a personalized diet, more attention is paid to the energy value of the diet and the ratio of the main components (proteins, fats and carbohydrates) depending on their share (in %) of the caloric content of the diet.

We agree with the comment made by this Reviewer. The present manuscript refers to standard portions of different food categories to assist both regulatory objectives and dietary guidelines for adults. Instead, this approach cannot be reliable for drawing personalized diets for weight control or other individual needs.

To improve the article Scientific Soundness, I recommend adding a comparison of the portion size of different food categories with their caloric content (add a table or chart).

We agree with the significant Reviewer's comment and suggestion, which would be helpful for a larger project at the European level to promote the determination of standard reference portions valid for all countries. However, this aim was different from our goal. Moreover, technically more limiting, no tables with the compositions of different foods are available for all countries. Thus, it is not easy to compare the calorie content of different food categories in diverse countries because there are quantitative and qualitative differences in the individual nutritional components for each category. Finally, although of marginal significance for the topic under discussion, after a detailed evaluation of the available data and related documents, we further selected the values to include in the tables and graphical representations. In particular, data related to Romania and Ukraine have been excluded: the available values referred to the total daily intake of a food category (e.g., 250 g of bread) or single intake units (much less than the portion), not to single portions.

Round 2

Reviewer 1 Report

Authors have carefully addressed the suggestions.

Reviewer 3 Report

After revision and correction of the comments, the article can be accepted for publication.